# Quantum enhanced radio detection and ranging with solid spins

Xiang-Dong Chen[1,2,3], En-Hui Wang[1,2], Long-Kun Shan[1,2], Shao-Chun Zhang[1,2], Ce Feng[1,2], Yu Zheng[1,2], Yang Dong[1,2], Guang-Can Guo[1,2,3] & Fang-Wen Sun [1,2,3] ✉

The accurate radio frequency (RF) ranging and localizing of objects has benefited the researches including autonomous driving, the Internet of Things, and manufacturing. Quantum receivers have been proposed to detect the radio signal with ability that can outperform conventional measurement. As one of the most promising candidates, solid spin shows superior robustness, high spatial resolution and miniaturization. However, challenges arise from the moderate response to a high frequency RF signal. Here, by exploiting the coherent interaction between quantum sensor and RF field, we demonstrate quantum enhanced radio detection and ranging. The RF magnetic sensitivity is improved by three orders to $21\,\mathrm{pT}/\sqrt{\mathrm{Hz}}$, based on nanoscale quantum sensing and RF focusing. Further enhancing the response of spins to the target's position through multi-photon excitation, a ranging accuracy of 16 $\mu$m is realized with a GHz RF signal. The results pave the way for exploring quantum enhanced radar and communications with solid spins.

The detection of electromagnetic field scattering and absorption is an indispensable tool for research from fundamental physics to industry applications, including remote sensing[1,2], imaging[3], communication[4] and ranging[5]. In free space, the diffraction determines the distribution of electromagnetic field, and limits the spatial resolution of imaging and precision of position estimation. Super-resolution optical far-field microscopy has been developed to beat the diffraction limit[6–9]. One of the most widely studied solutions is based on the nonlinear interaction with a structured light field[10–12]. Multi-photon excitation, as a typical nonlinear process, has been proven to be a powerful tool for high resolution microscopy imaging. It can enhance the spatial resolution with a factor of approximate $\sqrt{N}$[13–15], where $N$ is the number of photons that are evolved in the optical process.

The idea of multi-photon interaction with a spatially modulated light field can be extended to the radio frequency (RF) regime, and help to develop high resolution RF imaging and ranging, which has attracted wide interest in autonomous driving[16], manufacturing[17], health monitoring[18] and the Internet of things[19,20]. However, the traditional multi-photon process is usually observed with a high power

excitation. It hinders applications with a free space RF field. A potential solution is to utilize the quantum properties of interaction between nanoscale system and a weak electromagnetic field[21–23]. The coherent interaction between quantum emitter and the light field is recently applied for high resolution optical imaging with a low excitation power[24,25]. Quantum illumination is also studied to improve the signal-to-noise ratio of imaging beyond the classical limit of $\sqrt{N}$[26,27].

In this work, by utilizing the coherent interaction between the RF field and solid state spins, we demonstrate quantum-enhanced radio detection and ranging under ambient conditions. The spin-RF interaction is enhanced approximately four orders of magnitude through the RF field focusing and quantum sensing at the nanoscale. It enables the detection of a free space GHz radio signal with a sensitivity of 21 $\mathrm{pT}/\sqrt{\mathrm{Hz}}$. An RF interferometer with the quantum sensor is subsequently built to detect the phase of a radio signal that is reflected by a target. The coherent multi-photon interaction improves the accuracy of radio ranging beyond the classical limit. The target's position is estimated with a precision of 16 μm (corresponding to $1.6 \times 10^{-4}\lambda$). The proof-of-principle results demonstrate the potential of solid spin

[1]CAS Key Laboratory of Quantum Information, School of Physical Sciences, University of Science and Technology of China, Hefei 230026, P. R. China. [2]CAS Center For Excellence in Quantum Information and Quantum Physics, University of Science and Technology of China, Hefei 230026, P. R. China. [3]Hefei National Laboratory, University of Science and Technology of China, Hefei 230088, P. R. China. ✉e-mail: fwsun@ustc.edu.cn

qubits in quantum radar, microwave photonics and high sensitivity metrology in astronomy.

## Results

### Design of the experiment

Phase estimation can provide a higher spatial resolution for radio ranging in comparison with the signal strength and time of flight methods. Here, the quantum sensor detects the phase of a backscattered RF signal through an RF interferometer. The concept is shown in Fig. 1a. The signal from a RF generator is split into two individual paths, which are subsequently radiated into free space by horn antennas. One path serves as the reference. In the second path, the free space RF signal is reflected by an aluminum plate ($40 \times 40$ cm$^2$). The phase difference between the backscattered and reference RF is then determined by the target's distance $L$ as:

$$\varphi = 4\pi \cdot \frac{L}{\lambda}, \tag{1}$$

where $\lambda$ is the wavelength of RF field in free space. Assuming the same magnetic amplitude ($B_1$) of the two paths (this is not a stringent requirement), the total amplitude of the interference in free space will vary as:

$$B_{RF} = 2B_1 \cdot |\cos\frac{\varphi}{2}|. \tag{2}$$

Then, the radio ranging is realized by detecting the interaction between the quantum sensor and a spatially modulated RF field (Fig. 1b). The resolution of ranging and imaging will be improved by enhancing the RF field sensitivity and exploring the nonlinearity of coherent multi-photon RF-spin interaction.

### Detection of radio signal

The quantum sensor of RF field is nitrogen vacancy (NV) color center ensemble (with a density of approximately 5000/μm$^2$) in diamond. It is produced through nitrogen implantation and subsequent annealing. The ground state of NV center is a spin triplet. The resonant frequency of the transition between $m_s = 0$ and $m_s = \pm 1$ is $\omega_\pm = D \pm \gamma B_z$, where $D = 2.87$ GHz is the zero-field splitting, $\gamma = 2.8$ MHz/G is the gyromagnetic ratio of electron spin and $B_z$ is an external static magnetic field along the axis of NV center. Therefore, the spin state transition of NV center will provide the information of a gigahertz RF signal. Here, the frequency of the carrier wave is set to 2.885 GHz, resonant with $\omega_+$ under a bias magnetic field, as shown in Fig. 2a.

Started by initializing the NV center in $m_s = 0$ with a green laser pulse, the interaction between NV center and a resonant RF field is depicted as the Rabi oscillation:

$$\rho_{|0\rangle} = \frac{1}{2}(1 + \cos(\Omega t_{RF})), \tag{3}$$

where $\rho_{|0\rangle}$ is the population of $m_s = 0$ state and $t_{RF}$ is the duration of RF pulse. The Rabi oscillation frequency is determined by the magnetic component of local RF field $B_{loc}$, as $\Omega = 2\pi\gamma B_{loc}$. The amplitude of local RF field is assumed to be linearly dependent on the RF field in free space as $B_{loc} = kB_{RF}$, with a conversion efficiency of $k$. In practice, the spin state transition is affected by the decoherence and inhomogeneous broadening with NV center ensemble. By detecting the spin-dependent fluorescence emission, we observe that the oscillation's amplitude exponentially decays with a time $\tau \approx 460$ ns, as shown in Fig. 2b. According to the results of Rabi oscillation, the free space RF field magnetic sensitivity with NV center ensemble is deduced as

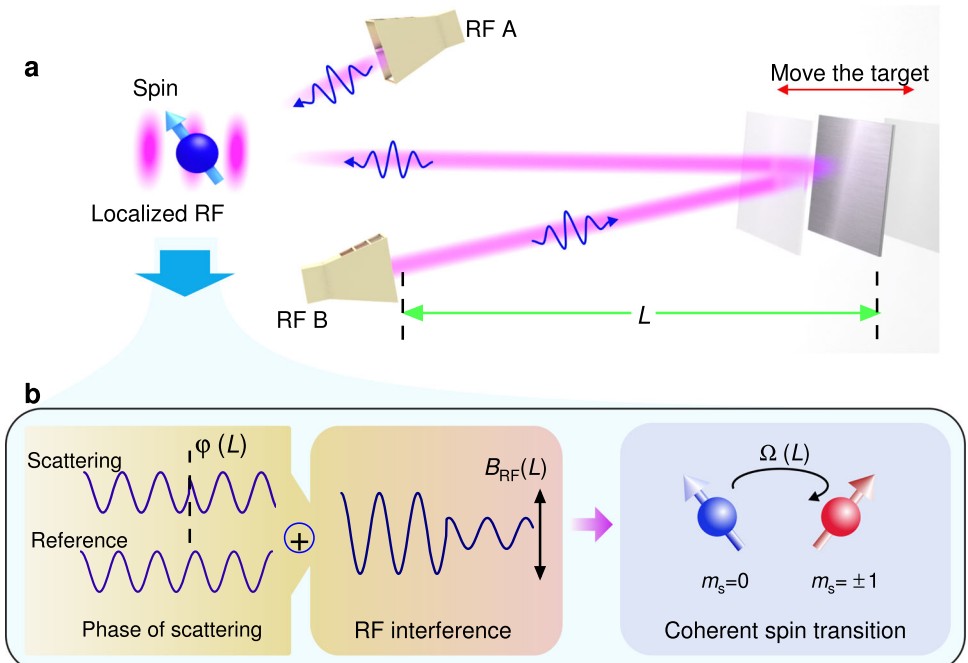

**Fig. 1 | The scheme of high accuracy RF ranging with a quantum sensor.** **a** Conceptual setup for the radio ranging with quantum sensor. Two RF paths with the same frequency serve as the reference (RF A) and ranging signal (RF B). The ranging signal is reflected by a target with a distance of $L$. Then, the free space interference signal between the two paths is confined in a microscale volume and interacts with the NV center quantum sensor. **b** Principle of extracting the target's distance information. The phase of the back scattered RF pulse changes with the position of the target, as $\varphi(L)$. It determines the amplitude ($B_{RF}$) of the interference between the backscattered and the reference RF pulses. Subsequently, the Rabi oscillation rate of quantum sensor will change with the position of target, as $\Omega(L)$. The position of target is finally estimated by measuring the electron spin of NV center ensemble in diamond.

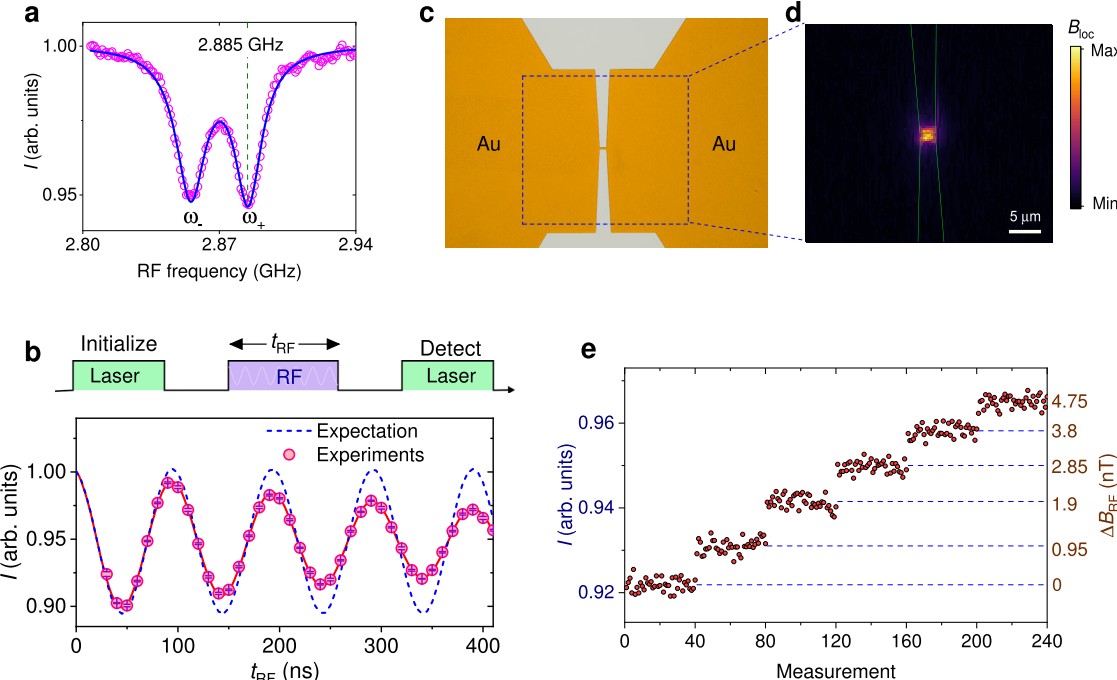

**Fig. 2 | The quantum sensing of free space RF signal. a** Optically detected magnetic resonance (ODMR) spectrum of NV center ensemble. The circles are experimental results. The solid line is the Lorentz fit. **b** The Rabi oscillation is visualized by initialization and detecting the spin state with a green laser (0.7 mW power). The pulse duration is $\frac{t_{det}}{2}$ (= 550 ns) for both the initializing and detecting. The dashed line indicates a perfect spin transition without decoherence and inhomogeneous broadening. High fluorescence emission means the spin state of $m_s = 0$, while low emission rate represents the spin state of $m_s = \pm 1$. The error bars represent the standard error of measurements. **c** The photograph illustrated the center of nanowire-bowtie structure. **d** The distribution of RF field, which is imaged by detecting the spin state transition under continuous-wave RF and laser excitation. **e** Real time measurement of RF amplitude modulation with NV center ensemble. The amplitude of RF source is changed with a step of 0.95 nT. Each experimental data point is obtained by repeating the spin detection pulse sequence $2 \times 10^5$ times. Only the reference RF is turned on here, and the power is approximately 2.3 mW.

(Supplementary Note 1):

$$\eta_B \propto \frac{1}{\gamma k C \sqrt{\epsilon}} \frac{\sqrt{1 + t_{RF}/t_{det}}}{t_{RF} e^{-t_{RF}/\tau}}. \tag{4}$$

Here, $C$ denotes the optical contrast between different spin states. $\epsilon$ consists of the photon emission rate of NV center ensemble and the fluorescence detection efficiency of setup. $t_{det}$ is the signal's detection time.

For the sensing of a weak radio signal, the idea is to focus the free space RF field and detect the spin state transition of NV center ensemble in a nano/micro scale volume. As shown in Fig. 2c, a nanowire-bowtie antenna on the diamond surface is adopted[23] (for details, see Supplementary Note 2). The two arms of the metallic bowtie structure are connected by a conductive wire with a width of approximately 1 $\mu$m. The antenna collects the free space RF signal and transmits the RF current through the nanowire (Fig. 2d). As a result, the magnetic component of local RF field and the interaction with NV center ensemble are enhanced by $7.6 \times 10^3$ times near the nanowire. Therefore, an excessive sensing volume, which includes the NV center away from the wire, should be avoided because it will bring a high level of useless background fluorescence signal. Here, using a confocal microscope, the fluorescence signal of NV center ensemble is detected with a spatial resolution of approximate 500 nm. This enable us to separate the spin state information of NV center ensemble in the strong interaction area. In this way, we increase $k$ to $7.6 \times 10^3$ while maintaining the properties of NV center sensor in Eq. (4).

The sensitivity of free space radio detection is experimentally estimated by comparing the noise ratio with the optical response to the RF field. In Fig. 2e, we record the fluorescence emission of NV center ensemble while changing the amplitude of RF field. With an RF-spin interaction time $t_{RF} = 265$ ns, which is close to $\tau$, the result shows that the normalized optical response is $\Delta I/\Delta B_{RF} \approx 1.1\%$/nT. Here, the fluorescence collection efficiency $\epsilon$ has been reduced 10 times to avoid the possible saturation of optical detector (Supplementary Note 3). Under this condition, the standard deviation of the experimental data with a total measurement time of 0.27 s is 0.14%. The RF field magnetic sensitivity is 66 pT/$\sqrt{Hz}$ with attenuated fluorescence signal. It corresponds to a sensitivity of 21 pT/$\sqrt{Hz}$ with all the fluorescence signal. Therefore, we estimate that a free space RF field electrical sensitivity of 63 $\mu$V/cm/$\sqrt{Hz}$ can be realized. It worth noting that, the decay time[28] and amplitude of Rabi oscillation with NV center ensemble change with the Rabi frequency (Supplementary Note 4). As a result, the RF sensitivity would be lower with a weaker RF signal. In future applications, this problem can be avoided by applying a strong bias RF field.

## Quantum enhanced radio ranging

During Rabi oscillation, a single RF $\pi$ pulse could pump the spin state from $m_s = 0$ to $m_s = \pm 1$, after the electron spin is initialized in $m_s = 0$. And a second $\pi$ pulse will pump the spin back to $m_s = 0$. This transition can be endlessly pumped with the sequential RF $\pi$ pulses, as long as the spin state coherence is preserved. It indicates that, the spin state Rabi oscillation with a multi-$\pi$ pulse can be seen as a coherent multi-photon process, with real intermediate states. In analogy to the multi-photon microscopy, Rabi oscillation will increase the spatial frequency of NV center ensemble's response to the spatially modulated RF field in Eq. (2). It subsequently improves the resolution of radio ranging with NV center ensemble.

Due to the large enhancement of local spin-RF interaction strength, the coherent multi-photon process can be pumped with a weak free space RF field here. To demonstrate the high resolution radio ranging with NV center ensemble, a peak power of 0.03 W at RF A

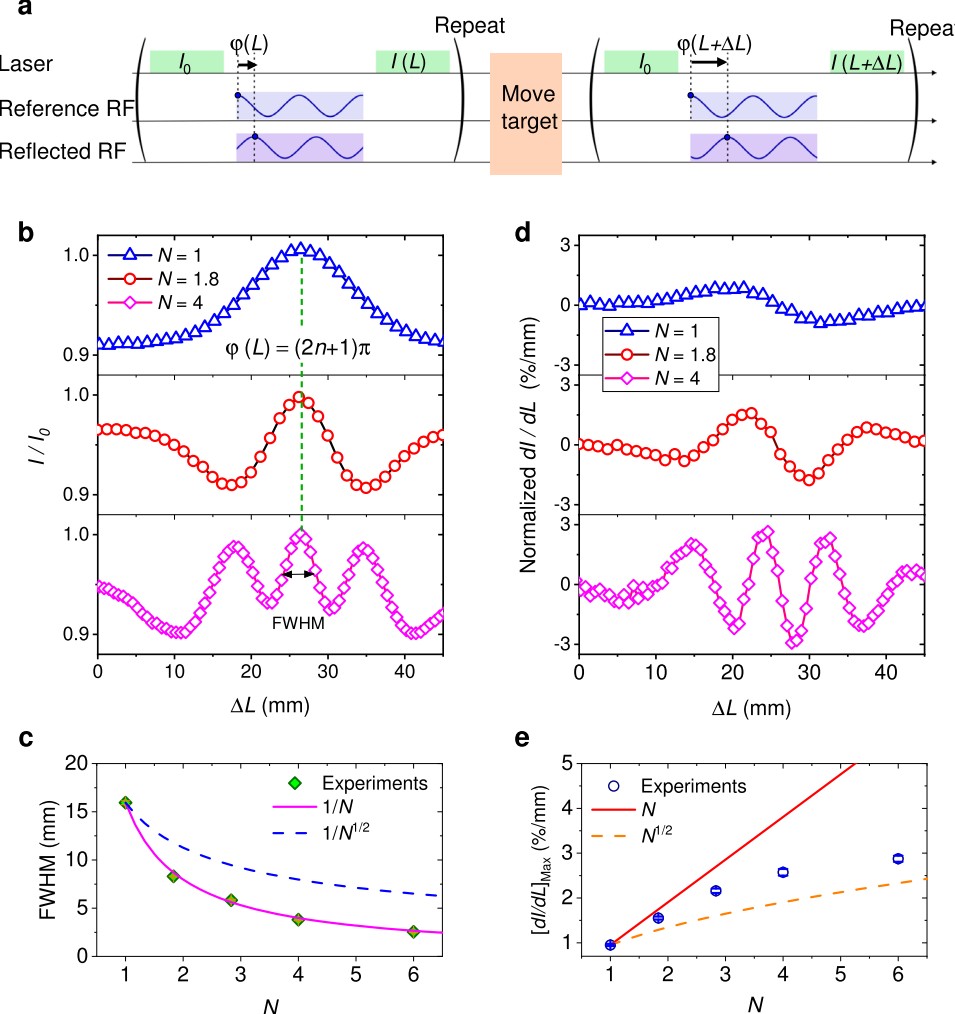

**Fig. 3 | The high accuracy radio ranging with quantum sensor. a** The pulse sequence for RF ranging. Fluorescence signals of NV center ensemble before and after the RF-spin interaction are recorded as a reference $I_0$ and a ranging signal $I(L)$. **b** The normalized ranging signal is plotted as the function of target's position. The number $N$ is modified by changing the RF pulse duration here. The smallest FWHM is obtained at the position where $\varphi(L) = \pi$ ($B_{RF} = 0$). **c** FWHM of the ranging signal is reduced as $\frac{1}{N}$. The green squares are experimental results, while solid and dashed lines represent the estimation of $1/N$ and $1/\sqrt{N}$, respectively. **d** The optical response of NV center ensemble to the target's position is deduced from the data in (**b**). **e** The maximum optical response $dI/dL$ increases with the RF $\pi$ pulse number beyond $\sqrt{N}$, due to the coherence of RF-spin interaction. The blue circles are experimental results, while solid and dashed lines represent the estimation of $N$ and $\sqrt{N}$, respectively. The data of optical response in (**d**, **e**) have been normalized by $I_0$. Error bars in (**c**, **e**) represent the standard error of several measurements.

and a peak power of 0.35 W at RF B are applied to balance the amplitude of reference and back scattered RF. The information of target's position is obtained with the pulse sequence in Fig. 3a. The spin state of NV center ensemble is initialized to $m_s = 0$, and the fluorescence is simultaneously recorded as an optical reference $I_0$. The RF pulse is then radiated by the horn antenna and reflected by the target. The interference between the backscattered and reference RF determines the spin state of NV center ensemble. The distance dependent fluorescence of NV center ensemble, as $I(L)$, is finally pumped with another green laser pulse.

Moving the target at a distance of approximately 2 m by a stepper motor, the amplitude of RF field interference varies between $2B_1$ (78 nT here) and 0. We define the RF excitation with $t_{RF}B_1 = \frac{N}{4k\gamma}$ as an $N$-$\pi$ pulse. As shown in Fig. 3b, with a single $\pi$ pulse (30 ns duration) excitation, the spin state of NV center changes from $m_s = +1$ to $m_s = 0$ by moving the target's position from $L = \frac{n}{2}\lambda$ to $(\frac{n}{2} + \frac{1}{4})\lambda$, where $n$ presents the integer ambiguities of ranging.

As explained before, the increase of $N$ enhances the nonlinearity of response to the RF field. More fringes appear with a higher $N$ in Fig. 3b. The full width at half maximum (FWHM), which is usually used

to characterize the spatial resolution of imaging, decreases with $N$ (Fig. 3c). It matches the expectation in Eq. (2) and (3) (details in Supplementary Note 5). In comparison, with tradition optical multi-photon excitation, the FWHM of a scanning microscopy is reduced with $\sqrt{N}$. The reduction of FWHM can help to develop high spatial resolution RF imaging. In addition, since a perfect Rabi oscillation will show no saturation of excitation, the resolution of imaging is theoretically unlimited by increasing $N$.

The FWHM does not include the contrast between the maximum and minimum fluorescence signals of NV center ensemble, which is related to the amplitude of Rabi oscillation. Therefore, spin state decoherence and inhomogeneous broadening do not change the reduction in FWHM. However, they will affect the evaluation of NV center ensemble's response to the target's position. In Fig. 3d, we plot the derivative of the normalized NV center ensemble fluorescence to the position of aluminum plate. The response of NV changes with the target's position. With a single $\pi$ pulse excitation, the highest normalized $dI/dL$ is found to be 0.9%/mm. The ratio increases to approximately 2.6%/mm with $N = 4$. It confirms that the maximum of $dI/dL$ increases with the RF excitation above $\sqrt{N}$, as shown in Fig. 3e.

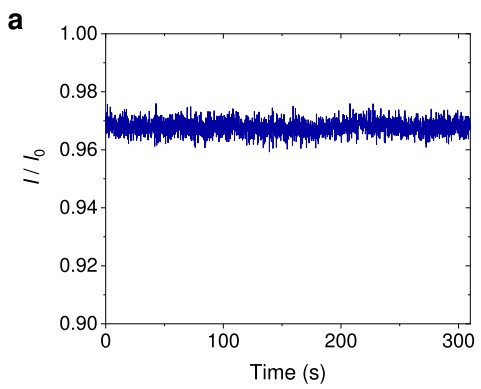
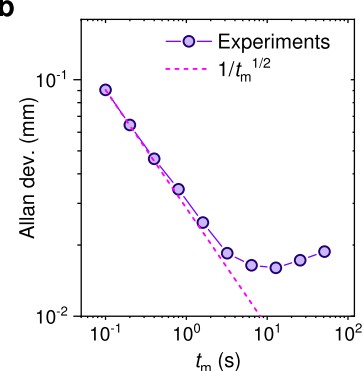

**Fig. 4 | The estimation of RF ranging accuracy. a** Time trace of the normalized ranging signal with $N=4$. The configuration of RF pulse is set to obtain a local maximum optical response $dI/dL$, as that in Fig. 3d. **b** Allan deviation of the data in (**a**) decreases with the measurement time $t_m$. The deviation of ranging is calculated by dividing the deviation of fluorescence signal by the optical response $dI/dL$. Circles are experimental results, while dashed line represents the shot-noise limit of $1/\sqrt{t_m}$.

However, the increase of optical response with a higher $N$ will be limited by the decoherence and inhomogeneous broadening with NV center ensemble, as explained before. In the experiments, we mainly test the ranging with $N\leq6$, as it is close to the highest expected accuracy. The RF phase sensitivity and ranging accuracy can be calculated from the optical response and the noise ($\sigma_I$) of fluorescence signal, as $\frac{\sigma_I}{dI/dL}$. Since the spin-RF interaction time ($30 \times N$ ns) is much shorter than the time for spin initialization and detection ($>1\mu s$) in the ranging experiments, the increase in $N$ does not substantially change the duty ratio of fluorescence detection. The noise ratio of ranging signal is almost the same with varied $N$. Therefore, the phase sensitivity and ranging accuracy is mainly determined by the normalized optical response, and will be enhanced beyond the classical limit of $\frac{1}{\sqrt{N}}$.

In Fig. 4a, we continuously record the RF ranging signal of NV center ensemble with $N=4$. The normalized noise ratio is $\frac{\sigma_I}{I_0} \approx 0.077\%$ with a one second measurement time. Compared with the optical response $dI/dL$, the accuracy of ranging with $N=4$ is estimated to be approximately $30\,\mu m$ with a one second measurement. It corresponds to a phase sensitivity of $4 \times 10^{-3} rad/\sqrt{Hz}$. In comparison, a phase sensitivity of $0.095 rad/\sqrt{Hz}$ was previously reported by performing high frequency quantum heterodyne with NV center[29].

To characterize the stability of the setup, the Allan deviation of RF ranging signal is analyzed in Fig. 4b. It shows that the highest ranging accuracy of $16\,\mu m$ can be obtained with a 10 s measurement time. The instability of experimental setup mainly results from the vibration of the building and shift of the sensor. Since the fluorescence collection of NV center ensemble has been reduced approximately 10 times here, this accuracy is expected to be achieved with a much higher sampling rate when collecting all the fluorescence signals.

The integer ambiguities of ranging based on phase estimation can be solved by applying frequency modulation. For example, an external magnetic field is applied to split the spin state transition frequencies of $\omega_\pm$. According to Eq. (1), the maximum unambiguous range ($R_{max}$) with single RF ($\omega_+$) satisfies $\Delta\varphi = 4\pi\frac{R_{max}\omega_+}{c} = 2\pi$, where $c$ is the speed of light. It is deduced that $R_{max} = \frac{c}{2\omega_+}$. In contrast, simultaneously detecting the NV center spin state transition with $\omega_+$ and $\omega_-$, the maximum unambiguous range will be extended as $4\pi\frac{R_{max}}{c}(\omega_+ - \omega_-) = 2\pi$. It indicates that $R_{max}$ can be extended by a factor of $\frac{\omega_+}{\omega_+ - \omega_-}$. In addition, the time of flight ranging can be performed with the quantum sensor(Supplementary Note 6). It provides the information on the target's position with a longer dynamic range, though a lower ranging accuracy.

For the ranging with a moving object, the Doppler shift of reflected signal is $f_d = 2v/\lambda$, where $v$ is the speed of object. Considering a subsonic speed, $f_d$ will be smaller than 10 kHz with RF pulse ($\lambda \approx 0.1$ m) in this experiments. In contrast, the width of ODMR peak (Fig. 1a) is greater than 10 MHz. Therefore, the Doppler shift of RF frequency will not change the amplitude of Rabi oscillation. We expect that the change of sensitivity can be ignored with normal moving objects.

## Discussion

In recent decade, quantum illumination[30–33] and quantum detector[34–38] have been proposed to improve the performance of a radar system. Comparing with other quantum RF receivers, such as Rydberg atoms and superconductors (where sophisticated setups or low temperature are needed), the NV center in diamond shows the advantages of robustness, high spatial resolution and miniaturization[39–43]. It is operated with simple setup under ambient condition. The working frequency band can be adjusted through Zeeman effect to tens of GHz[44,45]. And the CMOS-compatibility of diamond is favored for the development of a compact and integrated quantum sensor[46,47]. Recently, heterodyne sensing with NV center has been demonstrated for the high resolution spectroscopy of GHz radio signal[29,48,49]. It can be used to analyze the Doppler shift of RF scattering with moving objects. Combining these techniques, we expect that the NV center can be used to develop a practical high resolution quantum radar.

The protocol of radio detection in this work improves the sensitivity of an ultra high frequency signal with NV center by approximately three orders[29,48]. It enables the prototype of NV-based high resolution radio ranging (with 0.35 W RF peak power) to realize an accuracy that is comparable with the harmonic radio-frequency identification system[19], which is operated with an RF power lower than 1 W. Meanwhile, Rydberg atoms RF receiver was recently demonstrated with electrical sensitivity from tens of $\mu V/cm/\sqrt{Hz}$ to sub-hundred $nV/cm/\sqrt{Hz}$[37,38,50], though it has not been applied for high resolution radio ranging. Considering the advantages of NV center RF detector, further enhancing the sensitivity of RF detection 1–2 orders would be helpful for improving the competitiveness of NV-based quantum radar in future applications. Potential approaches include optimizing the NV center production, spin manipulation and increasing the fluorescence collection and RF focusing efficiency. For example, the inhomogeneous broadening with NV center ensemble, which limits the optical contrast $C$ and the decay time $\tau$, can be reduced by optimizing the diamond engineering[40]. Using high density NV center ensemble in $^{12}C$ isotopic purified diamond may further improve the RF field sensitivity by three orders[51,52]. In addition, optimizing the RF antenna[53] and utilizing spin-wave in ferromagnetic materials[54,55] will provide a solution for further enhancing of local RF-spin interaction, increasing $k$ in Eq. (4).

In conclusion, we have studied high resolution radio ranging with a solid state quantum sensor under ambient conditions. An RF field magnetic sensitivity of $21\ pT/\sqrt{Hz}$, and a radio ranging accuracy of $16\,\mu m$ is demonstrated with quantum sensing. With efforts to further

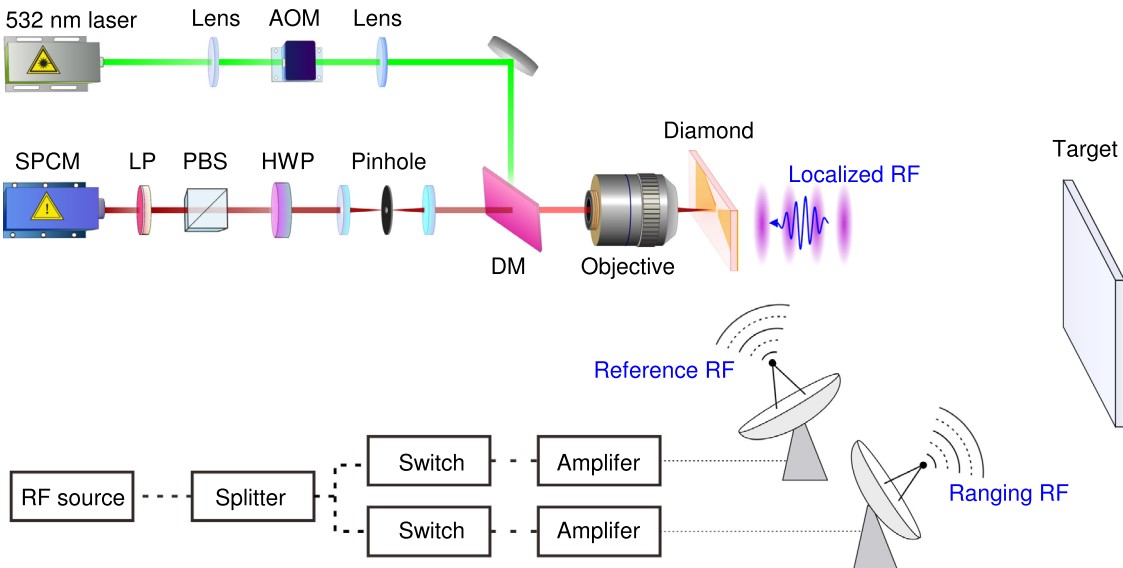

**Fig. 5 | The schematic diagram of experimental setup for the radio ranging with NV center.** DM, long-pass dichroic mirror with an edge wavelength of 552 nm; AOM acousto-optic modulator, SPCM single-photon-counting modulator, PBS polarizing beam splitter, LP longpass filter with an edge wavelength of 633 nm; HWP half-wave plate.

enhance the sensitivity and dynamic range, we hope that quantum enhanced radio detection and ranging can be used for research on autonomous driving, navigation, health monitoring and wireless communication.

## Methods

### Sample preparation

The diamond plate is a $2 \times 2 \times 0.5$ mm$^3$ size single crystal with (100) surface from Element 6. High density NV center ensemble is produced through nitrogen ion implanting (15 keV energy, $10^{13}$/cm$^2$ dosage) and subsequent annealing (850 °C for 2 h). After the production of NV center ensemble, a nanowire-bowtie structure of chromium/gold (5/200 nm thickness) film is produced on the diamond surface through lift-off. And it is subsequently connected to a cm size in-plane bowtie antenna, which is made with copper foil tape.

### Experimental setup

The NV center ensemble is optically detected with a home-built confocal microscope, as shown in Fig. 5. A 532 nm laser (MGL-DS-532nm, New Industries Optoelectronics) is used to excite the spin-dependent emission of NV center, and is modulated by an acousto-optic modulator (MT200-0.5-VIS, AA). An objective with 0.7 NA (Leica) focuses the laser onto the diamond and collects the fluorescence of NV center ensemble. The fluorescence is detected by a single-photon-counting-module (SPCM-AQRH-15-FC, Excelitas) after passing through a long-pass filter (edge wavelength 633 nm, Semrock).The diamond quantum sensor is mounted on a glass plate, which is moved by a three-dimensional manual stage. The distance between diamond and the objective is fine tuned by a piezo stage (NFL5DP20/M, Thorlabs), while the direction of laser beam is scanned by a Galvo mirror system (GVS012/M, Thorlabs).

The RF from a signal generator (SMA100A, Rhode&Schwartz) is split into two paths by a power splitter (ZFRSC-42-S, MiniCircuits). RF pulse of each path is subsequently controlled by switches. The two paths are separately amplified by two amplifiers (60S1G4A, Amplifier Research, and ZHL-16W-43+, MiniCircuits). The RF of two paths is radiated into free space by two horn antennas (LB-2080-SF, Chengdu AINFO Inc). One path of RF is reflected by the target, the position of which is moved by a stepped motor. The other path directly pumps the NV center.

## Reporting summary

Further information on research design is available in the Nature Portfolio Reporting Summary linked to this article.

## Data availability

Additional data are available from the corresponding author upon reasonable request. Source data are provided with this paper.

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

## Acknowledgements

This work was supported by the Innovation Program for Quantum Science and Technology (No. 2021ZD0303200); CAS Project for Young Scientists in Basic Research (No. YSBR-049); National Natural Science Foundation of China (No. 62225506); Key Research and Development Program of Anhui Province (No.2022b13020006); and the Fundamental Research Funds for the Central Universities. The sample preparation was partially conducted at the USTC Center for Micro and Nanoscale Research and Fabrication.

## Author contributions

X.C. and F.S. conceived the idea. X.C. performed the experiments. E.W. and C.F. prepared the samples. L.S. and Y.Z. built the electrical setup. X.C., S.Z., Y.D. and F.S. analyzed the data. All authors contributed to the discussion and editing of manuscript.

## Competing interests

The authors declare no competing interests.
