## [Peer Review File · Nature Communications]

REVIEWER COMMENTS

Reviewer #1 (Remarks to the Author):

In this manuscript Chen et al. demonstrate the detection of RF field using an ensemble of NV centers. The authors use a gold antenna fabricated with lithography on the diamond surface to concentrate the magnetic field around the NVs, which allows them to reach a sensitivity of 21 pT/VHz. While this value is lower than reports from the literature showing sensitivity around 1 pT/VHz (ref. 44 in the manuscript), the method described here has a potential and can be further improved. Additionally, the authors demonstrate ranging, a kind of “radar” application, with a spatial accuracy of 16 μm . I believe the results reported here could be interesting for the quantum sensing community and could recommend the manuscript for publication in “Nature Communications” when the points below addressed:

1. The authors use an ensemble of NV centers as a quantum sensor, but it is often referred in the text as “Nitrogen Vacancy color center” singular (page 2, section “Detection of radio signal” and other places in the text). This is confusing, since the reader might think that a single NV center is used for the experiments. I would recommend the authors to change the text accordingly.
2. The RF amplitude is denoted as B_0 in the text, but in the literature it is common to use B_1 .
3. In the caption of figure 1 the authors write that the “RF field is confined in a nanoscale volume”, but according to the simulations shown in figure 2 c, the actual volume is several cubic micrometers. Please correct this in the revised manuscript.
4. On page 3, section “Quantum enhanced radio ranging” the authors claim, that “The transition of spin state during Rabi oscillation can be seen as a coherent multi-photon process...”, but they do not explain why. Please elaborate more here.
5. The authors defined a RF photon number $N=4k\gamma t_{\text{RF}} B_0$, but this is misleading since there are no single Rf photons detected in the experiment. This equation actually gives the number of the Rabi oscillations, so another definition should be use to avoid confusion.

6. Page 5, end of left column. The authors write that the FWHM has an $1/N$ dependence, which is “matching the expectations from eq. 2 and 3. This effect does not follow immediately from these equation, so a more detail explanation is needed here.

7. Page 5, end of left column. The authors claim that “Therefore, the phase sensitivity and ranging accuracy will be enhanced beyond the classical limit of \sqrt{N} ”, but I think a better explanation is needed, why this is the case.

Reviewer #2 (Remarks to the Author):

The manuscript by Chen et al. describes a novel scheme for detecting the range of the objects using an interferometry scheme in microwave domain. The

microwave photon phase estimation scheme is utilised with a quantum microwave receiver as NV center in diamond at room temperature.

Authors present a proof of principle experiment as well as the idea and experimental proof of quantum advantage of non linear (multiphoton) absorption of the photons by the detector and hence a more sensitive response on the phase, compared to the single photon absorption.

The usage of NV for detecting the MW is new. This is a noteworthy achievement, including the MW antenna for focusing the MW field. As regards to the other works in the field the comparison to other detector types is lacking.

Paper also lacking references on Quantum Radar topic which was recently outlined

(<https://www.science.org/doi/10.1126/sciadv.abb0451>,

<https://journals.aps.org/prl/abstract/10.1103/PhysRevLett.128.010501>,

<https://journals.aps.org/prl/abstract/10.1103/PhysRevLett.114.080503>,

<https://arxiv.org/pdf/2111.03409.pdf>)

To my knowledge, this is a first demonstration of quantum ranging platform, based on NV center, one of the candidates for the application, which operates in realistic scenario. In order to properly estimate the achievement, some quantitative comparison with other major competitors platform is

missing, which reduces the quality, as does not show the outlook and how much this platform still needs to improve to be competitive with other platforms.

The work is done on high technical level with usage of state of the art research methods in the field of quantum sensing with NV centers.

The data analysis and discussion is on high level.

To my taste the discussion of the manuscript could be improved.

In my opinion the paper would benefit reflection the following aspects and comments:

1. Discussion about the ranging is limited to resting objects, i.e. speed is 0.

As we know most of the objects are moving, what happens to the signal?

Will the method still work, what is the maximum speed at which it will work?

How the ranging sensitivity would reduces because of that?

2. Maybe I am getting it wrong, but I could not correlate the numbers related to sensitivity on page 3,

According to the main text, the $dI/dB = 1,1\%/nT$, the noise dI for 0,27s, is 0,14%, the sensitivity thus is $\eta = 0,14\% \cdot \sqrt{0,27 \text{ s}} / 1,1\% / nT = 66 \text{ pT}/\sqrt{\text{Hz}}$, please provide a clear calculation for the number 27 pT/sqrt(Hz), which is given in the text.

3. The $T_2\rho$ time (decay if the Rabi oscillations) depends on Rabi frequency as is known (ref <https://iopscience.iop.org/article/10.1088/1367-2630/abd2e5>). This should be accounted in the sensitivity curve (linear dependence of signal vs the distance, if you are not in the single working distance L)

4. The discussion about extending the dynamic range was a bit brief, the main idea of modulation could be open for more clarity.

5. As the ranging accuracy depends on the power in the MW (for shot noise limited detection), the power of MW should be used in the ranging accuracy and compared with other techniques

6. Last sentence of conclusion: This claim is too strong and maybe could be relaxed, since it was not shown directly in the manuscript

7. Further enhancing the sensitivity of RF detection is necessary for improving the competitiveness of solid spin quantum radar — how much is missing compared to other platforms?

8. How the $4 \cdot 10^{-3}$ rad compares to other methods of phase estimation (of microwaves)?

9. Why $N = 4$ was the maximum tested?

10. Why not use a bigger diamond sensing volume (authors say they used 500 nm confocal spot) , what is the optimum for ranging sensitivity?

11. How usage of other detectors of MW (Atoms, Superconducting qubits, classical microwave receivers) compares to the present work, is there additional benefit of NV platform compared to others, apart from Room temperature, and ambient conditions?

We thank the Editor and Reviewers for their helpful comments! We have modified the manuscript according to those comments. The changes in the manuscript are marked by blue color. The main changes are summarized as follows:

- 1, We emphasized that NV center ensemble was used in the work.
- 2, “B1”, instead of “B0”, was used to denote the amplitude of RF in the revised manuscript.
- 3, Fig. 2c was updated to show the distribution of local RF field more precisely.
- 4, More explanation of the comparison between Rabi oscillation and multi-photon process was added in the first paragraph of section “Quantum enhanced radio ranging”.
- 5, The changes of FWHM and ranging accuracy with N were explained clearer in the main text and supplementary information.
- 6, We added a discussion of ranging with a moving object.
- 7, More explanations about the RF magnetic sensitivity estimation were added.
- 8, The results of Rabi oscillation with different RF powers were added in Supplementary Information.
- 9, We added more discussion about extending the dynamic range.
- 10, We added the comparison of RF power with traditional radar system in the second paragraph of “Discussion” section.
- 11, More explanations about the maximum tested N were added.
- 12, More discussions about further enhancing sensitivity were added.
- 13, The reason for not using a very large sensing volume was explained.
- 14, More comparisons with other platforms were added.
- 15, Grammar errors were checked.
- 16, The format of the manuscript was further adjusted according to the requirements of the journal.

In the following, we will present the point-by-point responses to the comments from the Reviewers.

Reviewer 1

Comment 1 (general comment)

“In this manuscript Chen et al. demonstrate the detection of RF field using an ensemble of NV centers. The authors use a gold antenna fabricated with lithography on the diamond surface to concentrate the magnetic field around the NVs, which allows them to reach a sensitivity of $21 \text{ pT}/\sqrt{\text{Hz}}$. While this value is lower than reports from the literature showing sensitivity around $1 \text{ pT}/\sqrt{\text{Hz}}$ (ref. 44 in the manuscript), the method described here has a potential and can be further improved. Additionally, the authors demonstrate ranging, a kind of “radar” application, with a spatial accuracy of $16 \text{ }\mu\text{m}$. I believe the results reported here could be interesting for the quantum sensing community and could recommend the manuscript for publication in “Nature Communications” when the points below addressed”

Our reply:

We thank the Reviewer for the positive and useful comments! We carefully modified the manuscript according to these suggestions.

Our work demonstrated the potential of using NV center and other solid state quantum system to realize high accuracy radio detection and ranging. Various applications can be explored based on this work. And it can be combined with other techniques to further improve the performance of radio detection and ranging. For example, in the following works, we plan to optimize the spin state manipulation of NV ensemble to improve the spin coherent interaction time. We estimate that the sensitivity can be improved 4-5 times by reducing the impact of inhomogeneous broadening. We are also planning to utilize the light guiding effect of silver nanowire to improve the efficiency of fluorescence collecting. We believe our method will realize a sensitivity much higher than $3.4 \text{ pT}/\sqrt{\text{Hz}}$, which was shown with a better diamond sample in ref. 52 (ref. 44 of previous version manuscript. “ $1 \text{ pT}/\sqrt{\text{Hz}}$ ” was the introduction of low frequency magnetic field sensitivity, not the RF sensitivity in the reference).

Comment 2

“The authors use an ensemble of NV centers as a quantum sensor, but it is often referred in the text as “Nitrogen Vacancy color center” singular (page 2, section “Detection of radio signal” and other places in the text). This is confusing, since the reader might think that a single NV center is used for the experiments. I would recommend the authors to change the text accordingly.”

Our reply:

We replaced presentation with “NV center ensemble”, at the places where we described the process of sensing.

Comment 3

“The RF amplitude is denoted as B_0 in the text, but in the literature it is common to use

B1.”

Our reply:

Thanks for the reminding! We used “B1” to denote the RF amplitude in the revised manuscript.

Comment 4

“In the caption of figure 1 the authors write that the “RF field is confined in a nanoscale volume”, but according to the simulations shown in figure 2 c, the actual volume is several cubic micrometers. Please correct this in the revised manuscript.”

Our reply:

We corrected the description in the revised manuscript. We also updated the image of local RF distribution in Fig. 2c. Previous image directly used the optically detected magnetic resonance (ODMR) contrast to represent the strength of local RF. The revised image was plotted by further considering the saturation of ODMR contrast under CW RF pumping. It would show the distribution of local RF signal more precisely than previous version.

Comment 5

“On page 3, section “Quantum enhanced radio ranging” the authors claim, that “The transition of spin state during Rabi oscillation can be seen as a coherent multi-photon process...”, but they do not explain why. Please elaborate more here.”

Our reply:

In the manuscript, in order to explain how the Rabi oscillation would increase the spatial frequency of ranging signal, we compared it to the multiphoton process, which has been used to improve the resolution of optical microscopy. Multiphoton excitation can be viewed as the transition through virtual or real intermediate states, as illustrated in Fig. R1.1a. In analogy to multiphoton excitation, during the Rabi oscillation, a single RF π pulse could pump the spin state transition from $m_s=0$ to $m_s = \pm 1$, after the spin is initialized in $m_s=0$. A subsequent second π pulse will pump the spin back to $m_s=0$. The transitions can be endlessly pumped as long as the spin state coherence is preserved. In each spin-flip, a single RF photon is absorbed or emitted. Therefore, the Rabi oscillation can be seen as a coherent multiphoton process with real intermediate states, as in Fig. R1.1b. The nonlinearity will then enhance the response to the modulation of RF excitation, similar to that for high resolution optical nonlinear microscopy.

We modified the sentences in the first paragraph of section “quantum enhanced radio ranging” to better explain the Rabi oscillation and multiphoton excitation.

Figure R1.1 Comparison between multiphoton excitation and Rabi oscillation.

Comment 6

“The authors defined a RF photon number $N=4k\gamma t_{RF} B_0$, but this is misleading since there are no single Rf photons detected in the experiment. This equation actually gives the number of the Rabi oscillations, so another definition should be use to avoid confusion.”

Our reply:

We apologize for the misleading! Here, we attempted to explain the nonlinear response of the ranging signal, by comparing the Rabi oscillation with an $N\text{-}\pi$ pulse to an N -photon process. The RF pulse with $t_{RF}B_1 = \frac{N}{4k\gamma}$ will coherently pump N cycles of spin flip between $m_s=0$ to $m_s=\pm 1$, similar to an N -photon process. It does not mean that the RF pulse only contains N photons. In the revised manuscript, the sentence was changed to “We define the RF excitation with $t_{RF}B_1 = \frac{N}{4k\gamma}$ as an $N\text{-}\pi$ pulse.” We also changed the similar presentation at other places.

Comment 7

“Page 5, end of left column. The authors write that the FWHM has an $1/N$ dependence, which is “matching the expectations from eq. 2 and 3. This effect does not follow immediately from these equation, so a more detail explanation is needed here.”

Our reply:

Thanks for the suggestion! We added more details of the derivation in the fourth section of revised Supplementary Information.

The full width at half maximum (FWHM) can be derived from the distance between the positions of maximum and half-maximum fluorescence signals. According to Eq.3

in the main text, the maximum NV center fluorescence signal is obtained at $\Omega t_{RF} = 0$, and the half maximum is obtained at $\Omega t_{RF} = \pi/2$. Considering Eq.2 of main text, we deduce that the position of target is $L_1 = \frac{1}{2}(n + \frac{1}{2})\lambda$ for $\Omega t_{RF} = 0$, where n is an integer. And the position L_2 for $\Omega t_{RF} = \pi/2$ is determined by $|\cos(2\pi L_2/\lambda)| = \frac{1}{2} \cdot \frac{1}{4\gamma k B_1 t_{RF}}$. Writing $L_2 = \Delta L + L_1$ and $4\gamma k B_1 t_{RF} = N$, we obtain

$$\left| \cos\left(\frac{2\pi\Delta L}{\lambda} + n\pi + \frac{1}{2}\pi\right) \right| = \frac{1}{2N}.$$

It can be written as $\sin\left(\frac{2\pi\Delta L}{\lambda}\right) = \pm \frac{1}{2N}$. In our experiment, $N \geq 1$ and $\Delta L \ll \lambda$, the equation will be approximated as

$$\frac{2\pi\Delta L}{\lambda} \approx \pm \frac{1}{2N}.$$

Then, the FWHM is deduced as $FWHM = 2|\Delta L| = \frac{\lambda}{2N\pi}$.

Comment 8

“Page 5, end of left column. The authors claim that “Therefore, the phase sensitivity and ranging accuracy will be enhanced beyond the classical limit of \sqrt{N} ”, but I think a better explanation is needed, why this is the case”

Our reply:

We added more explanation in the revised manuscript.

With a given measurement time, the minimum detectable change in the target's position (σ_L) is determined by the response of NV centers to the target's position (dI/dL) and the noise (σ_I) of NV fluorescence signal, as $\sigma_L = \frac{\sigma_I}{dI/dL}$. The noise ratio changes with the duty ratio of fluorescence detection pulse in each measurement cycle, as $\frac{\sigma_I}{I_0} \propto \frac{1}{\sqrt{t_{det}/t_{all}}}$, where t_{all} includes both the RF and laser pulse durations. In this work, the RF pulse duration is $30N$ ns, much shorter than the laser pulse (>1 μ s). The noise ratio $\frac{\sigma_I}{I_0}$ changes by less than 5% when N changes from 1 to 4. Therefore, the phase sensitivity and ranging accuracy will be mainly determined by the normalized optical response $\frac{dI/dL}{I_0}$. In the experiments, we found that the optical response of NV center was increased above $N^{1/2}$, due to the coherent interaction. As a result, the ranging accuracy will be enhanced beyond $1/N^{1/2}$.

Reviewer 2

Comment 1 (general comment)

“The manuscript by Chen et al. describes a novel scheme for detecting the range of the objects using an interferometry scheme in microwave domain. The microwave photon phase estimation scheme is utilised with a quantum microwave receiver as NV center in diamond at room temperature. Authors present a proof of principle experiment as well as the idea and experimental proof of quantum advantage of non linear (multiphoton) absorption of the photons by the detector and hence a more sensitive response on the phase, compared to the single photon absorption.

The usage of NV for detecting the MW is new. This is a noteworthy achievement, including the MW antenna for focusing the MW field. As regards to the other works in the field the comparison to other detector types is lacking. Paper also lacking references on Quantum Radar topic which was recently outlined (<https://www.science.org/doi/10.1126/sciadv.abb0451>, <https://journals.aps.org/prl/abstract/10.1103/PhysRevLett.128.010501>, <https://journals.aps.org/prl/abstract/10.1103/PhysRevLett.114.080503>, <https://arxiv.org/pdf/2111.03409.pdf>)

To my knowledge, this is a first demonstration of quantum ranging platform, based on NV center, one of the candidates for the application, which operates in realistic scenario. In order to properly estimate the achievement, some quantitative comparison with other major competitors platform is missing, which reduces the quality, as does not show the outlook and how much this platform still needs to improve to be competitive with other platforms.

The work is done on high technical level with usage of state of the art research methods in the field of quantum sensing with NV centers.

The data analysis and discussion is on high level.

To my taste the discussion of the manuscript could be improved.”

Our reply:

In general, we appreciate the positive comments and useful suggestions from the Reviewer. In this revised manuscript, we modified the presentation of manuscript according to the comments from the Reviewer. In particular, we added more references including those mentioned by the Reviewer. More discussion about the comparison with other works (such as Rydberg atoms) was also added. Details of the replies to specific comments are shown below.

Comment 2

“Discussion about the ranging is limited to resting objects, i.e. speed is 0. As we know most of the objects are moving, what happens to the signal? Will the method still work, what is the maximum speed at which it will work? How the ranging sensitivity would reduce because of that?”

Our reply:

Thanks for the good question!

For moving objects, the Doppler shift of the reflected signal is $f_d = \frac{2v}{\lambda}$ (v speed of object; λ wavelength of RF signal). For the RF used in our experiments ($\lambda \approx 0.1m$), the Doppler shift with normal objects (such as cars) will be smaller than 10 kHz. Meanwhile, the spectral width of ODMR signal (Fig.2a in main text) is greater than 10 MHz with NV center ensemble in our experiments. Therefore, the change of RF frequency with a scale of 10 kHz will not change the amplitude of Rabi oscillation, as shown in Fig. R2.1 (where RF frequency changes by 0.5 MHz). We expect that the change in sensitivity can be ignored with normal moving objects. We will explore the method for detecting the Doppler shift of a moving object in future studies.

Figure R2.1 The Rabi oscillation of NV center ensemble with different RF frequencies.

To evaluate the response of NV center to a moving object, we write the phase of reflecting signal as $\varphi_{reflect} = 2\pi(f_0 + f_d)t - 2\pi \frac{2L_0}{\lambda} + \varphi_0$, where L_0 is the initial distance and φ_0 is the initial phase. The phase of reference signal is written as $\varphi_{reference} = 2\pi f_0 t + \varphi_0$. For the ranging protocol in this work, the fluorescence of NV center depends on the phase difference between reference and reflecting RF signals as $\Delta\varphi = 2\pi \frac{2L_0}{\lambda} - 2\pi f_d t = 2\pi \frac{2}{\lambda}(L_0 - vt)$. We can see that the signal will change with the real time position of object.

In the last paragraph of section “Quantum enhanced radio ranging”, we added the discussion of ranging with a moving object.

Comment 3

“Maybe I am getting it wrong, but I could not correlate the numbers related to sensitivity on page 3, According to the main text, the $dI/dB = 1,1\%/nT$, the noise dI for 0,27s, is 0,14%, the sensitivity thus is $\eta = 0,14\% \cdot \sqrt{0,27 \text{ s}} / 1,1 \% / nT = 66 \text{ pT}/\sqrt{\text{Hz}}$, please provide a clear calculation for the number 27 pT/sqrt(Hz), which

is given in the text.”

Our reply:

Sorry for the confusion! We added more explanation about the sensitivity in the revised manuscript.

The collected fluorescence intensity changes with the RF pulse duty ratio in the experiments. A high fluorescence intensity with a short RF duration might lead to saturation of the single photon counting module (SPCM). Therefore, to compare the sensitivity with different RF pulse durations, we inserted a neutral density (ND) filter with OD = 1 (reduces fluorescence 10 times) in the fluorescence collecting path for most of the measurements, as explained in the manuscript. With the reduced fluorescence intensity, the direct sensitivity is $66pT/\sqrt{Hz}$, as pointed out by the reviewer. Since the noise ratio decreases with the square root of the total fluorescence intensity, the sensitivity of this protocol will reach $\left(\frac{66}{\sqrt{10}}\right)pT/\sqrt{Hz} \approx 21pT/\sqrt{Hz}$ by collecting all the fluorescence signals.

Comment 4

“The T2rho time (decay if the Rabi oscillations) depends on Rabi frequency as is known (ref <https://iopscience.iop.org/article/10.1088/1367-2630/abd2e5>). This should be accounted in the sensitivity curve (linear dependence of signal vs the distance, if you are not in the single working distance L)”

Our reply:

Thank you for the careful reading!

The Reviewer is correct that the Rabi oscillation decay time changes with Rabi frequency. As shown below in Fig. R2.2, we measured the Rabi oscillation of NV center in our diamond sample under different RF excitations. We observed that the decay time decreased with Rabi frequency. Meanwhile, the amplitude of Rabi oscillation increased with Rabi frequency, due to inhomogeneous broadening (Fig. R2.2b). Overall, the response of NV center ensemble to RF magnetic field will increase with the Rabi frequency (as in Fig. R2.2c).

Other effects, such as the nonlinearity with Rabi oscillation, also contribute to the variation of sensitivity with different target positions. In fact, as in Fig. 3d of manuscript, we have shown that the ranging sensitivity of NV center changes with the target's position.

Figure R2.2 **The RF detection changes with the RF power.** **a** The Rabi oscillation of NV center ensemble with different excitations. **b** The decay time (τ) and amplitude (ΔI) of oscillation changes with Rabi frequency. **c** The optical response of NV center with different Rabi frequencies. The RF pulse duration time is set to 200 ns in (c).

To solve this problem, one solution is to enhance the RF magnetic sensitivity with a weak RF signal. Methods include applying a strong bias RF field. Thus, radio detection will work in the high sensitivity regime. Another potential solution is applying RF frequency modulation. Since the RF spatial distribution changes with the frequency, we can adjust the sensitivity at certain positions by changing RF frequency.

In the revised manuscript, we added the explanation about position-dependent sensitivity in the last paragraph of section "Detection of radio signal" and in the fifth paragraph of "Quantum enhanced radio ranging". The results of Rabi oscillation with different RF powers were added in Supplementary Information. The reference that is mentioned by the Reviewer has been added.

Comment 5

"The discussion about extending the dynamic range was a bit brief, the main idea of modulation could be open for more clarity."

Our reply:

We added more discussion about extending the dynamic range.

Frequency modulation is widely used for continuous wave radar. For the ranging with a single frequency RF, the reflected signal can be written as $B_1 \sin(2\pi\omega t + \varphi)$, where the phase $\varphi = \frac{4\pi L}{\lambda}$ (as Eq. 1 in the main text). The maximum unambiguous range (L_{\max}) is determined as $\Delta\varphi = 2\pi$. Therefore, we derive that $L_{\max} = \frac{\lambda}{2} = \frac{c}{2\omega}$ (ω is set as ω_+ in the manuscript).

Considering the ranging with two RF signals (with frequency $\omega_+, \omega_- \gg \omega_+ - \omega_-$), the maximum unambiguous range should satisfy $\Delta\varphi_+ = n \times 2\pi$ and $\Delta\varphi_- = (n - 1) \times 2\pi$, where n is an integer. The maximum unambiguous range is derived as $\Delta\varphi_+ - \Delta\varphi_- = \frac{4\pi L_{\max}}{c}(\omega_+ - \omega_-) = 2\pi$. In this condition, we obtain $L_{\max} = \frac{c}{2(\omega_+ - \omega_-)}$.

Therefore, by utilizing two frequencies for ranging, we can extend the maximum unambiguous range by a factor of $\frac{\omega_+}{\omega_+ - \omega_-}$.

Comment 6

“As the ranging accuracy depends on the power in the MW (for shot noise limited detection), the power of MW should be used in the ranging accuracy and compared with other techniques”

Our reply:

We added the comparison of RF power with traditional ranging system in the second paragraph of “Discussion” section.

As explained in the manuscript, we applied an RF peak power of 0.35 W for ranging. A ranging accuracy of 30 μm is demonstrated by detecting 10% of the total fluorescence signal in one second. In comparison, the authors in Ref.19 (Nat. Electron. 2, 125 (2019)) performed harmonic radio-frequency identification ranging with UHF band RF power less than 1 W. They realized a ranging 50 μm accuracy. It shows that the accuracy of our work is at the same level as the state-of-the-art classical ranging system. The sampling rate should be improved by approximately 1 order of magnitude to enhance the competitiveness. As discussed in the manuscript, potential methods include optimizing diamond sample preparation and spin manipulation, increasing fluorescence collection efficiency. For example, we are planning to test silver nanowire for transmitting both optical and electrical signals. It can increase the sensing volume of NV center, and subsequently enhance the fluorescence intensity. We expect that the sampling rate of this quantum radar can be significantly improved in future studies.

Comment 7

“Last sentence of conclusion: This claim is too strong and maybe could be relaxed, since it was not shown directly in the manuscript”

Our reply:

Thank you for the suggestion! We changed the sentence to “With efforts to further enhance the sensitivity and dynamic range, we hope that quantum enhanced radio detection and ranging can be used for research on autonomous driving, navigation, health monitoring and wireless communication.”

Comment 8

“Further enhancing the sensitivity of RF detection is necessary for improving the competitiveness of solid spin quantum radar — how much is missing compared to other platforms?”

Our reply:

We added clearer discussion in the revised manuscript.

With stable fluorescence and long spin coherence time at room temperature, NV center in diamond has been extensively studied as quantum sensors in condensed matter physics, biology and earth science. Recently, it was proposed to detect the high frequency RF signals. In this work, we utilized a nanowire-bowtie structure to enhance the high frequency RF sensitivity more than 3 orders of magnitude, and subsequently demonstrated a micrometer scale ranging accuracy with NV center. As explained in the reply of comment 6, a further $\sqrt{10}$ times sensitivity enhancement will help to outperform the harmonic radio-frequency identification ranging system. For the purpose of communication and RF detection, Rydberg atom RF receiver, with electrical sensitivity from tens of $\mu\text{V}/\text{cm}/\text{Hz}^{1/2}$ to sub-hundred $\text{nV}/\text{cm}/\text{Hz}^{1/2}$, was recently demonstrated (Nature 535, 262–265 (2016); AIP Adv. 9, 045030 (2019); Nat. Phys. 16, 911–915 (2020)). This indicates that the sensitivity of NV center should be enhanced more than 2 orders of magnitude to reach the highest-level sensitivity of Rydberg atoms.

Considering the advantages of robustness, high integration and high spatial resolution, we expect that a further 1-2 order enhancement of sensitivity will help to promote the applications of NV-based high resolution radio ranging, which was not demonstrated with Rydberg atoms. Enhancing the sensitivity with NV center is one of the most important tasks in our following works. In addition to the design of new nanostructures (as in the reply to comment 6), we are also planning to optimize spin manipulation with NV center ensemble. For example, we estimate that the sensitivity can be enhanced 4-5 times by reducing the impact of inhomogeneous broadening. In addition, it has been shown that the isotopically purified diamond can increase the spin

dephasing time 1-2 orders (Nat. Commun. 10,3766(2019)). Therefore, we believe that the 2 orders enhancement of RF sensitivity is achievable with further studies.

Comment 9

“How the $4 \cdot 10^{-3}$ rad compares to other methods of phase estimation (of microwaves)?”

Our reply:

We added the comparison in the revised manuscript. In ref. 29 (Phys. Rev. A 104, L020602(2021)), the authors demonstrated the sensing of 1.51 GHz microwave with the technique of high-frequency quantum heterodyne with NV center. A phase sensitivity of $0.095 \text{ rad/Hz}^{1/2}$ was achieved with a microwave power at the same level as that in this work. Note that, we are detecting a reflected signal in this work, in contrast to the direct microwave pumping in ref.29.

Comment 10

“Why $N = 4$ was the maximum tested?”

Our reply:

The amplitude of Rabi oscillation decays with the RF duration, due to the decoherence and inhomogeneous broadening of NV center ensemble. As a result, the increase of ranging accuracy will slow down with a high N . In this work, we have performed the radio ranging with $N=6$ (results not shown in the previous manuscript). The ranging accuracy improves only 10% in comparison with $N=4$. It is close to the expected best accuracy. Therefore, we tested ranging with $N \leq 6$ in the experiments.

In the revised manuscript, we updated Fig. 3 to include the results with $N=6$. And we added more explanation about the maximum tested N .

Comment 11

“Why not use a bigger diamond sensing volume (authors say they used 500 nm confocal spot) , what is the optimum for ranging sensitivity?”

Our reply:

Thanks for the suggestion! Using a large volume of NV centers to improve the fluorescence intensity is one important method to enhance the sensitivity. In this work, we used a nanowire-bowtie structure to collect the free space RF signal and enhance the local RF-spin interaction. RF field is mainly confined around the wire (1 μm width), as shown in Fig. 2c. For the sensing of an RF signal, only the fluorescence of NV center near the wire is useful. Therefore, a confocal microscope is used to selectively collect the fluorescence of NV center ensemble. The resolution of 500 nm, which is determined by the NA of the available objective, is close to the width of RF field distribution. It balances the collection efficiency and spatial selection. An excessive sensing volume will bring a high level of background. As suggested by the Reviewer, we will try to use

a large volume of NV for sensing in the future to enhance the sensitivity. Possible solutions include utilizing spin-wave to extend the area of strong RF-spin interaction.

In the revised manuscript, we explained the reason for not using a bigger sensing volume in this work.

Comment 12

“How usage of other detectors of MW (Atoms, Superconducting qubits, classical microwave receivers) compares to the present work, is there additional benefit of NV platform compared to others, apart from Room temperature, and ambient conditions?”

Our reply:

Thanks for the comment! We added more discussions about the usage of solid spin sensor in the revised manuscript.

Compared with traditional electrical microwave sensors, no electrical amplifier is needed for NV center and other quantum sensors. The reduction of electrical components will reduce the electrical noise, which limits the performance of classical detectors. In addition, the resonant frequency of NV can be changed through Zeeman effect, making it able to detect signals of a wide frequency band (DC to tens even hundreds of GHz, Commun. Eng. 1, 19 (2022)).

As pointed out by the Reviewer, compared with other quantum detectors, the NV center quantum sensor can be operated under ambient and extreme conditions, even with high temperature (1000 kelvin in Nat. Commun. 10, 1344(2019)) and high pressure (GPa in Science 366, 1349(2019)). It enables the NV center quantum sensor to be used in various environments. Apart from room temperature and ambient conditions, NV center detector also shows other advantages, such as

- (1) High reliability. The high performance of NV center detector can be realized with low requirement of experimental setup. For example, lasers with narrow linewidth are needed for Rydberg atoms receiver. In contrast, a free-running diode laser is sufficient for NV center ranging in this experiment.
- (2) High integration. In this work, we build the entire prototype ranging system on a small aluminum profile trolley. It can be further miniaturized. Recently, on-chip (Nat. Electron. 2, 284 (2019)) and fiber coupling (Adv. Quantum Technol. 4, 2000111 (2021)) NV center sensor have been demonstrated. This implies a high level of integration and stability in future applications with NV center RF detector.
- (3) High spatial resolution. Due to the small size of NV center, it is able to detect the signal's distribution with high spatial resolution. In this work, by detecting the distribution of RF field, we have demonstrated the radio ranging with resolution of micrometers. Here, the lateral resolution is limited by the size of the nanowire-bowtie structure. We should balance the sensitivity and lateral resolution by using different sizes of nanowire-bowtie structure for future applications.

These advantages of solid spin quantum detector make it more practical than other quantum systems for applications. Currently, the sensitivity of NV detector is lower than that of Rydberg atoms. It shows the potential to be significantly enhanced with further studies, as discussed in the manuscript.

REVIEWERS' COMMENTS

Reviewer #1 (Remarks to the Author):

The authors have responded to all critics raised during the first review. The manuscript is significantly improved and now I can recommend it for publication in "Nature Communications".

Reviewer #2 (Remarks to the Author):

Authors have thoroughly replied to all my comments,

With changes made I recommend this paper for publication Nature communication